# Physiological Demands of a Self-Paced Firefighter Air-Management Course and Determination of Work Efficiency

**DOI:** 10.3390/jfmk8010021

**Published:** 2023-02-06

**Authors:** Andrew R. Jagim, Joel A. Luedke, Ward C. Dobbs, Thomas Almonroeder, Adam Markert, Annette Zapp, Andrew T. Askow, Richard M. Kesler, Jennifer B. Fields, Margaret T. Jones, Jacob L. Erickson

**Affiliations:** 1Sports Medicine, Mayo Clinic Health System, La Crosse, WI 54650, USA; 2Exercise & Sport Science Department, University of Wisconsin—La Crosse, La Crosse, WI 54601, USA; 3College of Health Professions, Trine University, Angola, IN 46703, USA; 4La Crosse Fire Department, La Crosse, WI 54603, USA; 5Fire Rescue Wellness, Chicago, IL 60653, USA; 6Department of Kinesiology and Community Health, University of Illinois at Urbana-Champaign, Urbana, IL 61801, USA; 7Illinois Fire Service Institute, Champaign, IL 61820, USA; 8Exercise Science and Athletic Training, Springfield College, Springfield, MA 01109, USA; 9Patriot Performance Laboratory, Frank Pettrone Center for Sports Performance, George Mason University, Fairfax, VA 22030, USA

**Keywords:** firefighter, occupational performance, occupational demands

## Abstract

Firefighters often complete air management courses (AMC) to assess the ability to tolerate personal protective equipment, appropriately manage the breathing system and assess occupational performance. Little information is known relative to the physiological demands of AMCs, nor how to assess work efficiency in order to characterize occupational performance and evaluate progress. Purpose: To assess the physiological demands of an AMC and examine differences across BMI categories. A secondary aim was to develop an equation to assess work efficiency in firefighters. Methods: Fifty-seven firefighters (Women, n = 4; age: 37.2 ± 8.4 yr.; height: 182.0 ± 6.9 cm; body mass: 90.8 ± 13.1 kg; BMI: 27.8 ± 3.6 kg·m^−2^) completed an AMC per routine evaluation while wearing a department issued self-contained breathing apparatus and full protective gear. Course completion time, starting pounds per square inch (PSI) on the air cylinder, changes in PSI, and distance traveled were recorded. All firefighters were equipped with a wearable sensor integrated with a triaxial accelerometer and telemetry to assess movement kinematics, heart rate, energy expenditure, and training impulse. The AMC consisted of an initial section involving a hose line advance, rescue (body drag), stair climb, ladder raise, and forcible entry. This section was followed by a repeating loop, which consisted of a stair climb, search, hoist, and recovery walk. Firefighters repeated the course loop until the self-contained breathing apparatus air supply pressure reached 200 PSI, at which time they were instructed to lay down until the PSI reached zero. Results: Average completion time was 22.8 ± 1.4 min, with a mean distance of 1.4 ± 0.3 km and an average velocity of 2.4 ± 1.2 m·s^−1^. Throughout the AMC, the mean heart rate was 158.7 ± 11.5 bpm equating to 86.8 ± 6.3% of the age-predicted max heart rate and a training impulse of 55 ± 3 AU. Mean energy expenditure was 464 ± 86 kcals and work efficiency was 49.8 ± 14.9 km·PSI^−1^·s. Regression analysis determined that fat-free mass index (R^2^ = 0.315; β = −5.069), body fat percentage (R^2^ = 0.139; β = −0.853), fat-free mass (R^2^ = 0.176; β = −0.744), weight (R^2^ = 0.329; β = −0.681), and age (R^2^ = 0.096; β = −0.571) were significant predictors of work efficiency. Conclusions: The AMC is a highly aerobic task with near-maximal heart rates reached throughout the course. Smaller and leaner individuals achieved a higher degree of work efficiency during the AMC.

## 1. Introduction

Firefighting is a physically demanding occupation with multiple known physical and environmental stressors, that place a high degree of cardiovascular and thermoregulatory strain on the body [1]. These physiological demands are exacerbated when wearing personal protective equipment and when utilizing self-contained breathing apparatuses (SCBA) [2,3,4]. As a result of these physiological demands, improving the fitness levels of firefighters has become a focal point within the United States. Further, a higher level of aerobic fitness, muscular endurance, and power has been shown to be associated with higher ratings of occupational performance, specifically air ventilation efficiency [1,5,6,7,8,9]. Furthermore, higher aerobic fitness and physical activity levels have been shown to be inversely related to risk factors for cardiovascular disease [10], which is prevalent among the firefighting profession and emergency personnel [11,12,13]. In addition to cardiovascular disease, combating issues pertaining to weight status and obesity is an area of concern for the firefighting profession. Previous research among a cohort of nearly 500 career firefighters reported that 80% of firefighters were classified as overweight (Body mass index [BMI] > 25 kg·m^−2^) and 34% classified as obese (BMI > 30 kg·m^−2^) [14]. To address the occupational demands of the profession and lower the risk of cardiovascular disease, there is a growing effort [15,16,17] to seek opportunities to improve aerobic capacity levels and overall fitness status for firefighters. As such, some departments perform annual physical fitness testing as part of annual evaluations or prior to employment. Specifically, many departments have implemented the Candidate Physical Ability Test (CPAT) before entrance to the academy, frequently requiring firefighters to complete a Work Performance Examination (WPE) prior to initiation of employment as full-time career firefighters. However, currently, there are no established national fitness standards or required annual fitness testing for firefighters long-term. Therefore, it is at the discretion of each department to oversee the implementation of fitness testing and occupational performance assessments.

In order to assess the influence of improved fitness status on occupational performance, it is important to select appropriate assessments. Standard laboratory measures and protocols are common for the evaluation of health and fitness parameters such as aerobic capacity, cardiovascular function, and body composition. Such tests may lack ecological validity and may not be accessible to all firefighters. Thus, efforts have been made to identify field-based measures of aerobic capacity [9] and fitness performance in addition to more specific indices of occupational performance through the use of firefighter tasks or work simulations [1,4,7,18,19,20,21,22]. An air management course (AMC) is an assessment technique used to evaluate occupational performance among firefighters while also assessing the ability to tolerate personal protective equipment, and appropriately manage the breathing system. However, a challenge with AMCs and similar occupational performance assessments, is they are often not standardized; thereby making the assessment of physiological demands and the characterization of performance outcomes challenging. Moreover, the lack of a standardized method to assess and quantify occupational performance precludes comparisons among firefighters and departments as well as monitoring progress over time. As such, there is a continued need to assess the physiological demands of firefighter-specific tasks and for the development of an equation to quantify occupational performance, while considering multiple factors (i.e., distance traveled, heart rate response, air utilization, time to completion, etc.).

A primary aim of an AMC is to assess the ability to utilize air from a SCBA, in an effort to maximize work output on a finite amount of air. Recently, a novel formula was developed to assess work efficiency during a simulated fireground test with a fixed endpoint (e.g., a set amount of work to complete in time) [23]. This formula enables the quantification of occupation-specific performance and further examination of how the field-based measure of work efficiency is associated with measures of fitness status and occupational performance. One limitation of this formula is that it may only be applicable to tasks with a fixed end point and may not be a valid measure of work efficiency during open-ended tasks, such as an AMC, which is considered open-ended as firefighters complete the course as many times as possible until air supply becomes limited. The development of a field-based measure of work efficiency to characterize occupational performance during open-ended firefighter-specific occupational assessments is warranted.

Understanding the physiological demands of occupation-specific tasks can enable practitioners to focus fitness programming efforts toward maximizing work efficiency and occupation-specific task performance. The development of a field-based measure of work efficiency can quantify occupational performance and facilitate the compilation of rankings for individual firefighters and provide normative data. Further, a singular measure of work efficiency would permit the examination of relationships between fitness parameters and occupation performance. Therefore, the purpose of the current study was to quantify the physiological demands of an AMC and to create a modified version of a previously developed field-based measure of work efficiency in order to characterize occupational performance among firefighters during an open-ended task. A secondary aim was to evaluate differences in physiological demands and work efficiency between groups based on weight status and to examine relationships between work efficiency and the physical characteristics of firefighters.

## 2. Materials and Methods

### 2.1. Study Design

This retrospective cohort study of firefighters from the same department included two days of testing. On day one, each firefighter completed the AMC wearing full protective gear, their department-issued SCBA, and an activity monitor. Various internal (e.g., heart rate) and external (e.g., distance covered) load variables, time to completion and air usage metrics were recorded during the AMC from which a composite measure of work efficiency was created. On a separate day of testing, firefighters completed a battery of fitness tests including a body composition assessment, isometric mid-thigh pull, maximal grip strength testing, and a movement screen.

### 2.2. Participants

Fifty-seven firefighters (Women, n = 4; age: 37.2 ± 8.4 yr; height: 182.0 ± 6.9 cm; body mass: 90.8 ± 13.1 kg; BMI: 27.8 ± 3.6 kg·m^−2^) participated in the study. All participants signed an institutionally approved informed consent form in accordance with the University Human Subject Research Guidelines. Approval was received from the fire department to utilize performance times and data derived from the activity monitors.

### 2.3. Procedures

*Body Composition*: Body mass and height were assessed using a self-calibrating digital scale and stadiometer to the nearest 0.1 kg and 0.5 cm, respectively. Body composition was assessed using a multi-frequency bioelectrical impedance analysis device, the H20N scale (InBody Inc., Cerritos, CA, USA) to determine body fat percentage (BF%), fat mass (FM) and fat-free mass (FFM).

*Isometric Mid-Thigh Pull (IMTP)*: An IMTP test was used to assess lower body maximal strength [24]. For the IMTP, the firefighters pulled upward as forcefully as possible on a stationary bar located at their mid-thigh level (between their knee and hip) by generating force through their lower bodies. The bar height was adjusted so that the initial knee and hip angles were approximately 125° and 145°, respectively [25]. Force platforms positioned under the firefighters’ feet sampled ground reaction force data at 1000 Hz throughout the IMTP (Hawkins Dynamics, Westbrook, ME, USA). Firefighters completed three maximal effort attempts, with 2-min of rest in between attempts. The peak forces for the three attempts were averaged.

*Grip Strength*: Firefighters were instructed to wrap their entire hand around a handheld dynamometer to attain maximal force production, with their arm fully extended at 90° of shoulder flexion. Firefighters completed three maximal effort attempts, with 2-min of rest given between attempts. Total grip strength was calculated by taking the highest value attained for each hand and summing them together.

*Movement Efficiency Test*: Firefighters completed an automated movement efficiency test using a web-based application (Fusionetics^TM^). The movement efficiency test consisted of seven individual movement tasks: 2-Leg Squat, 2-Leg Squat with Heel Lift, 1-Leg Squat, Push-Up, Shoulder Movements, Trunk Movements, and Cervical Movements. A licensed athletic trainer evaluated each movement and scored it according to manufacturer guidelines. An overall movement efficiency score was then calculated within the web-based calculation using a range of 0–100 (worst–best) based on previously observed movement compensations associated with each movement [26].

#### 2.3.1. Air Management Course (AMC)

Firefighters completed the self-paced AMC wearing department-issued SCBA and full protective gear. A member of the research team followed each firefighter throughout the course to record data at specific checkpoints. The time of completion for each section and the total course were recorded in addition to changes in air pressure (pounds per square inch [PSI]) while also wearing the activity monitor. Firefighters continued throughout the course until reaching an air pressure of 200 PSI (self-monitored on SCBA), at which time the firefighter stopped and maintained a recovery position (of their choosing), with the goal of reducing ventilation rate and maximizing their time left on air. The firefighter continued in this position until an air pressure of 0 PSI was reached.

The AMC course consisted of an initial section involving a hose line advance, rescue (body drag), stair climb, ladder raise, and forcible entry. This section was followed by a repeating loop, which consisted of a stair climb, search, hoist, and recovery walk as detailed below (A schematic of the AMC can be found in Appendix A):


**Initial Section:**


*Hose line advance*: Firefighters advanced a 100 ft section of a charged 1 ¾” hose line over a distance of 30.5 m in a straight line before flowing water for 2 s.

*Rescue*: Firefighters grasped a mannequin (mass 50 kg, height: 180 cm) underneath the shoulders using a “seatbelt” grip (under the armpits) and dragged the mannequin 30.5 m in a backward direction.

*Stair Climb*: A dry hose line (mass: 10.2 kg) was packaged as a high-rise pack and placed over the back of the SCBA. The firefighter carried the hose and a tool bag (mass: 5 kg) up three flights of stairs. The high-rise pack and tool kit were placed at the landing of the fourth flight of stairs. The firefighter completed the standpipe connection before descending the stairs.

*Ladder Raise*: Firefighters grasped a 7.32 m extension ladder affixed to a wall and extended it to full length using a hand-over-hand technique. Firefighters then returned the ladder to the resting length using the same technique.

*Forcible Entry*: Firefighters struck a simulated forcible entry chopping device (Keiser FORCE Machine, Keiser Co., Fresno, CA , USA) 20 times using a 3.6 kg sledgehammer.


**Continuous Circuit:**


*Stairs*: Firefighters entered the tower and climbed two flights of stairs.

*Search*: Firefighters performed a left-hand search by crawling approximately 23 m in a serpentine pattern.

*Hoist*: Firefighters raised a dry rolled 2.5 hose line (10.2 kg) using a hand-over-hand grip technique approximately 10 m (two stories) off the ground before lowering it back using the same technique.

*Recovery Walk*: Firefighters completed a 60 m recovery walk before returning to the training building and starting another lap.

#### 2.3.2. Activity Monitors

All firefighters were equipped with a GPS-based monitoring system with built-in heart rate monitoring system capabilities (Polar TeamPro^TM^ Polar Electro, Oy, Finland). Demographic information, including age, height, and weight, was entered into the proprietary software program associated with the monitoring system, which was used to determine age-predicted maximum heart rate (APMHR). Heart rate zones were used to quantify intensity based on the following zones: zone 1 = 50–60%, zone 2 = 60–70%, zone 3 = 70–80%, zone 4 = 80–90%, zone 5 = 90–100% of APMHR. In addition, Training Load a proprietary metric calculated from HR intensity and task duration was collected to quantify internal strain. Additionally, training impulse (TRIMP) values were calculated using the Banister method. Energy expenditure was estimated using the software program associated with the activity monitors; derived from a proprietary algorithm. Following completion of the AMC, sensors were removed, placed into a docking station, and synced to a cloud-based software program. Data were then exported from this program and used for analysis.

A composite score to assess work efficiency was generated from the individual parameters collected during the AMC and modified from the formula developed by Norris et al. [23]. The formula below was used: $$\left[\left(\frac{\mathrm{Distance}}{\mathrm{Pre}\;\mathrm{AMC}\;\mathrm{Air}-\mathrm{Post}\;\mathrm{AMC}\;\mathrm{Air}}\right)\times \mathrm{Time}\right]\times 100$$

### 2.4. Statistical Analysis

Descriptive statistics were generated to summarize the physiological demands of the AMC. Data are presented as means ± standard deviations. Data were stratified by BMI status (> or <25 kg·m^−2^) and independent T-tests were utilized to determine if differences existed between groups based on BMI status. Simple linear regression analyses were conducted to examine the extent to which age, body mass, height, BMI, FFM, FFM index (FFMI), movement efficiency, grip strength, and lower body strength could predict work efficiency during the AMC. Correlation coefficients were calculated to assess relationships and interpreted as: very weak: <0.20, weak: 0.20–0.39, moderate: 0.40–0.59, strong: 0.60–0.79, or very strong: >0.80 [27]. Statistical significance was determined as *p* < 0.05. All data were analyzed using IBM SPSS Statistics for Windows (Version 26.0; IBM Corp. Armonk, NY, USA).

## 3. Results

A summary of body composition parameters, lower body strength, and movement efficiency scores is presented in Table 1. When stratified by BMI, those with a BMI < 25 kg·m^−2^ (n = 21) had a lower body mass (mean difference [95% confidence intervals]) (−16.7 [−22.4, −10.6] kg), BF% (−8.8 [−11.8, −5.7]%), less FFM (−6.4 [−10.9, −1.9] kg) and a lower FFMI (−2.1 [−2.9, −1.4] kg·m^−2^) compared to those with a BMI > 25 kg·m^−2^ (n = 36). No significant differences were observed for absolute measures of lower body strength, grip strength, and movement efficiency between groups. When normalized to body mass, those with a BMI < 25 kg·m^−2^ produced more force on the IMTP (4.85 [2.2, 7.5] N; *p* < 0.001) and grip strength (0.15 [0.037, 0.269] kg; *p* = 0.01) assessments, compared to those with a BMI > 25 kg·m^−2^.

For the AMC, firefighters completed an average of 5.7 ± 1.3 laps (25:10 mm:ss), corresponding to a total distance of 1.34 ± 0.23 km, with an energy expenditure of 465 ± 86 kcals and work efficiency value of 49.8 ± 14.9 km·PSI^−1^·s. When stratified by BMI, those with a BMI < 25 kg·m^−2^ completed more laps (1.79 [1.24, 2.35]), achieved a longer duration (272 [138, 405] s), and completed more distance (0.22 [0.160, 0.338] km). Those with a lower BMI had a lower rate of air usage (−0.39 [−0.61, −0.18] PSI∙s^−1^), yet expended more energy (51.4 [4.0, 98.9] kcals) compared to those with a BMI > 25 kg·m^−2^. A summary of the aforementioned variables is included in Table 2.

A summary of the physiological responses during the AMC is provided in Table 3. Throughout the AMC, the mean heart rate was 158.7 ± 11.5 bpm (86.8 ± 6.3% APMHR) and the peak heart rate achieved was 181.5 ± 10.2 bpm (98.9 ± 5.6% of APMHR). This resulted in a TRIMP value of 79.2 ± 26.3 AU. No differences in cardiovascular responses were observed between groups when stratified based on BMI status.

Regression analysis determined that and FFMI (R^2^ = 0.315; β = −5.069), BF% (R^2^ = 0.139; β = −0.853), FFM (R^2^ = 0.176; β = −0.744), body mass (R^2^ = 0.329; β = −0.681), and age (R^2^ = 0.096; β = −0.571) were significant predictors of work efficiency. There was a weak inverse relationship between BF% (r = −0.373) and work efficiency; a moderate, inverse relationship between body mass (r = −0.573), FFM (r = −0.420), FFMI (r = −0.561), age (r = −310) and work efficiency (*p* < 0.01). No other physical characteristics, measure of lower body strength, or movement efficiency were associated with work efficiency (*p* > 0.05). There was a weak inverse relationship (*p* < 0.001) between body mass and TRIMP scores (r = −0.364), a moderate inverse relationship between body mass and total distance (r = −0.439) and body mass and time on course (r = −0.577), and a strong inverse relationship between body mass and the number of laps completed (r = −0.711). There was a weak inverse relationship (*p* < 0.01) between BF% and total distance (r = −0.324) and time on course (r = −0.301), and a moderate inverse relationship between BF% and the number of laps completed. There was a moderate inverse relationship (*p* < 0.01) between FFM and TRIMP scores (r = −0.425), number of laps completed (r = −0.429), and time on course (r = −0.492). There was a weak inverse relationship (*p* < 0.01) between FFMI and total distance completed (r = −0.315), a moderate inverse relationship between FFMI and TIMIP scores (r = −0.445), and the number of laps completed (r = −0.518), and a strong inverse relationship between FFMI and time on course (r = −0.613).

## 4. Discussion

The primary aim of the current study was to quantify the physiological demands of a firefighter AMC and examine differences across BMI categories. The main findings indicate that the AMC is a highly aerobic task, as evidenced by the near maximal heart rates achieved during the course, and a mean heart rate that equated to nearly 87% of APMHR, and nearly half of the course time being spent in HR Zone 5 (>90% APMHR). This level of activity equated to a TRIMP score of 79 AU and an estimated energy expenditure of ~465 kcals in 22 min of activity (~21 kcals·min^−1^). Individuals with a lower weight relative to height tended to perform better during the AMC; those with a BMI < 25 kg·m^−2^ completed more laps, achieved a longer duration, and completed more total distance.

It is advantageous to develop strategies and identify metrics for the purpose of evaluating occupational performance and characterizing field-based measures of fitness among firefighters. While not a direct assessment of total work completed, the amount of distance covered could be used as a surrogate measure of exercise capacity and as an indication of the ability of a firefighter to complete more work on a finite amount (i.e., tank) of air, an outcome with a high degree of ecological validity. Further, an aim of the current study was to develop a modified version of the previously published formula for work efficiency in firefighters. The firefighters achieved a work efficiency of 49.8 ± 14.9 km·PSI^−1^·s using the current formula, which is similar by design to exercise economy, a metric commonly assessed in laboratory settings. Additionally, it appears individuals with a BMI < 25 kg·m^−2^ achieved a greater work efficiency during the AMC compared to individuals with a BMI > 25 kg·m^−2^. Those with a lower BMI also had a lower rate of air usage, likely contributing to higher work efficiency. Previous work has also explored various firefighter simulations to assess occupational performance. For example, a simulated fire ground test, which consists of seven tasks designed to simulate job demands, has been used to quantify occupational performance [4,19]. While similar in nature to the first section of the AMC from the current study, its duration was much shorter (~365–400 s time to completion); however, the mean heart rate response (87.5% of age-predicted max HR) was comparable to that from the current study. Similarly, Gendron et al. [5] employed a simulated work circuit to evaluate air ventilation efficiency in firefighters. The test was composed of 10 different tasks, completed without rest in a serial fashion and the findings indicated that firefighters who performed the work circuit faster had lower air cylinder ventilation values (r = −0.495) and a higher peak oxygen consumption rate (r = −0.924). The authors concluded that firefighters with a faster performance during a simulated work challenge (a closed task) had better air ventilation efficiency. In terms of practical applications, this would suggest that improved air ventilation efficiency could potentially prolong time on air for firefighters actively engaged in fireground tasks while breathing from their SCBA.

A secondary aim of the current study was to develop an equation to assess work efficiency in firefighters and examine predictors of performance on the AMC. It is important to note that in the current study, occupational performance was assessed using the modified equation for work efficiency. As a result of the variables included in the equation, and the highly aerobic nature of the AMC, findings indicate that younger age, lower body weight, BF%, FFM, and FFMI were significant predictors of work efficiency. Due to the aerobic nature of the AMC, it is expected that smaller, leaner individuals would perform better and exhibit a higher degree of work efficiency as they likely utilize a lower amount (and rate) of air to do the same amount of work as seen previously [28,29] and similar to what is observed with endurance athletes [30]. Conversely, previous studies have utilized different firefighter tasks or simulations to evaluate occupation-specific performance and identify predictors of performance with conflicting findings [7,8,16,17,18,20,23,31]. For example, occupational performance has been assessed using a hose pull, stair climb while carrying a high-rise hose pack, simulated victim drag, equipment hoist, and forcible entry with individual completion times for each task used as performance measures or summed together for an overall measure of job performance [7,8]. Findings from these studies have reported positive associations between various indices of upper and lower body strength, in addition to anaerobic capacity and occupational performance [7], which may be a function of the short duration and high intensity of such previously selected tasks. Therefore, depending on the specific tasks selected to assess occupational performance, there may be a bias towards more aerobic-based predictors of occupational performance (as seen with the current study) versus predictors more reflective of strength and power. Ultimately, these findings provide evidence in support of a comprehensive fitness training program for firefighters that includes a wide array of strength and aerobic training.

## 5. Limitations and Future Directions

This work is not without limitations. Direct assessment of ventilation and oxygen consumption were not available yet would have provided more direct measures of work efficiency and the metabolic cost of an activity. Similarly, the lack of a criterion measure of oxygen consumption or ventilation prohibited the ability to directly validate the current work efficiency formula. Another limitation of the current study is the proprietary nature of the AMC. While the AMC included individual tasks that are commonly completed during firefighter activities and within the literature to assess occupational performance, the entire course and nature of the repeating loop may not be used universally by firefighter departments. Therefore, the current work efficiency formula may not be appropriate for all occupational performance tasks. It is recommended that future work seek to create a standardized AMC (or similar firefighter challenge course), which would allow for improved transferability across the literature and better evaluation of performance predictors or the characterization of occupational performance across the field. We did not have control over the number of females participating in the study and did not have equal distribution, which precluded any comparisons by sex. Lastly, it is important to note that only a select number of fitness characteristics were evaluated for their ability to predict work efficiency. Other physical and fitness-related characteristics may be able to predict occupational performance. Similarly, in the current study, occupational performance was only assessed using a measure of work efficiency during the AMC; therefore, other physical and psychological attributes may serve as important indicators of occupational performance among firefighters. Future research should strive for a larger sample size that would allow for the examination of between-group differences in addition to a more robust profile of performance, anthropometric, and biomechanical parameters that may serve as predictors of occupational performance and work efficiency in firefighters across multiple indicators of performance evaluation.

## 6. Conclusions

The findings from the current study indicate that younger and smaller (or those with a lower BMI) firefighters tend to perform better on the AMC and achieve a higher degree of work efficiency. Moreover, the AMC appears to be a highly aerobic task, with near-maximal heart rates achieved throughout the course. The current measure of work efficiency provides the ability to characterize occupational performance, which allows for the compilation of rankings within a department or comparisons from year to year. Future research should examine additional predictors of performance on the AMC to inform fitness programming efforts.

## Figures and Tables

**Table 1 jfmk-08-00021-t001:** Descriptive statistics for body composition, strength, and movement efficiency.

Variable	BMI Category	Value	Lower	Upper
Age (yr)	<25	34.9 ± 9.1	30.8	39.1
	>25	38.6 ± 7.7	35.9	41.2
	All	37.2 ± 8.4	35.0	39.4
Body Mass (kg)	<25	80.3 ± 7.3 *	76.9	83.6
	>25	96.9 ± 11.8	92.9	100.9
	All	90.8 ± 13.1	87.3	94.3
Body Fat Percent (%)	<25	14.9 ± 4.7 *	12.7	17.1
	>25	23.7 ± 5.7	21.7	25.7
	All	20.4 ± 6.8	18.5	22.3
Fat-free Mass (kg)	<25	68.8 ± 7.8 *	65.1	72.4
	>25	75.2 ± 8.1	72.3	78.1
	All	72.8 ± 8.5	70.4	75.1
Fat-free Mass Index (kg·m^−2^)	<25	20.8 ± 1.4 *	20.1	21.4
	>25	22.9 ± 1.3	22.4	23.4
	All	22.1 ± 1.7	21.6	22.6
IMTP Peak Force (N)	<25	2886 ± 422	2683	3090
	>25	3044 ± 441	2884	3203
	All	2985 ± 437	2862	3108
Grip Strength (kg)	<25	109 ± 16	101	117
	>25	117 ± 19	110	124
	All	114 ± 18	109	119
Movement Efficiency (AU)	<25	76.4 ± 10.1	71.4	81.2
	>25	74.7 ± 9.9	71.2	78.3
	All	75.3 ± 9.9	72.5	78.1

Data presented as mean ± standard deviation and 95% confidence intervals * Denotes *p* < 0.05; IMTP = Isometric mid-thigh pull; AU = arbitrary units.

**Table 2 jfmk-08-00021-t002:** Summary of completion characteristics during the AMC.

Variable	BMI Category	Value	Lower	Upper
Laps (n)	<25	6.9 ± 1.2 *	6.5	7.4
	>25	5.1 ± 0.9	4.8	5.5
	All	5.7 ± 1.3	5.4	6.1
Change in Air (PSI)	<25	4182 ± 292	4047	4316
	>25	4087 ± 292	3988	4186
	All	4120 ± 293	4040	4200
Time on course (s)	<25	1686 ± 280 *	1579	1793
	>25	1414 ± 203	1335	1493
	All	1510 ± 265	1438	1582
Rate of Air Use (PSI·s^−1^)	<25	2.54 ± 0.41 *	2.36	2.71
	>25	2.93 ± 0.36	2.80	3.06
	All	2.79 ± 0.42	2.68	2.91
Total Distance (km)	<25	1.48 ± 0.21 *	1.39	1.58
	>25	1.26 ± 0.19	1.19	1.33
	All	1.34 ± 0.23	1.28	1.40
Energy Expenditure (kcals)	<25	498 ± 101 *	459.9	536.4
	>25	447 ± 72	418.6	474.9
	All	465 ± 86	441.4	488.3
Work Efficiency (km·PSI^−1^·s)	<25	60.6 ± 16.1 *	52.8	68.3
	>25	43.9 ± 10.4	40.4	47.5
	All	49.8 ± 14.9	45.7	53.9

Data presented as means ± standard deviation and 95% confidence intervals. * Denotes *p* < 0.05.

**Table 3 jfmk-08-00021-t003:** Summary of cardiovascular responses during the AMC.

Variable	BMI Category	Value	Lower	Upper
TRIMP (AU)	<25	87.5 ± 33.8	75.6	99.4
	>25	74.7 ± 20.4	65.9	83.5
	All	79.2 ± 26.3	72.06	86.42
Max HR (bpm)	<25	183.4 ± 11.0	178.7	188.0
	>25	179.6 ± 9.5	176.2	183.0
	All	181.5 ± 10.2	178.2	183.7
Max HR (%)	<25	99.1 ± 6.9	96.5	101.6
	>25	98.9 ± 4.8	96.9	100.8
	All	98.9 ± 5.6	97.4	100.5
Mean HR (bpm)	<25	160.1 ± 12.8	154.8	165.4
	>25	157.9 ± 10.8	154.0	161.8
	All	158.7 ± 11.5	155.6	161.8
Mean HR (%)	<25	86.5 ± 7.9	83.6	89.5
	>25	86.9 ± 5.3	84.8	89.1
	All	86.8 ± 6.3	85.1	88.5
% Time in HR Zone 1	<25	2.4 ± 3.8	0.5	4.3
	>25	2.0 ± 4.3	0.6	3.4
	All	2.2 ± 4.1	1.1	3.3
% Time in HR Zone 2	<25	3.9 ± 3.8	1.5	6.3
	>25	6.1 ± 5.9	4.3	7.9
	All	5.3 ± 5.3	3.9	6.8
% Time in HR Zone 3	<25	15.2 ± 14.5	10.5	19.9
	>25	9.6 ± 7.0	6.1	13.1
	All	11.6 ± 10.5	8.7	14.4
% Time in HR Zone 4	<25	34.4 ± 17.5	27.0	41.8
	>25	35.7 ± 15.2	30.3	41.2
	All	35.2 ± 15.9	30.9	39.6
% Time in HR Zone 5	<25	42.7 ± 31.9	30.8	54.7
	>25	46.3 ± 22.1	37.5	55.1
	All	45.1 ± 25.7	38.0	52.1

Data presented as means ± standard deviation and 95% confidence intervals. HR zone 1 = 50–60%, HR zone 2 = 60–70%, HR zone 3 = 70–80%, HR zone 4 = 80–90%, HR zone 5 = 90–100%.

## Data Availability

Data available upon request.

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
