# Peer review of "Physiological Demands of a Self-Paced Firefighter Air-Management Course and Determination of Work Efficiency"

_jfmk, 2023, doi:10.3390/jfmk8010021_

Round 1
Reviewer 1 Report
The purpose of the study was to assess the physiological demands of an air management course and examine differences across BMI categories. The study has sound methodology and the manuscript is generally well written. I only have a couple of questions that i hope the authors could address.
1. In line 147: Beckham et al. recommended 125° knee and 145° hip in the study you referenced. But you stated it the other way round “initial hip and 146 knee angles were approximately 125° and 145°, respectively.” Why is that so?
2. The current data included 4 females. Females generally have different body fat percentage as men. Hence, the BMI reading may not be applicable to men, and vice versa. Would the authors be able to provide another set of data that exclude the female participants? As a table in the supplementary document at least.
3. The current sample size of female participants is too small to allow for this data to be generalised for the female as well. Should state this in the limitations.
Author Response
Reviewer 1:
The purpose of the study was to assess the physiological demands of an air management course and examine differences across BMI categories. The study has sound methodology and the manuscript is generally well written. I only have a couple of questions that i hope the authors could address.
Author Response: Thank you for taking the time to review our work. Please see a point-by-point response to each revision below:
In line 147: Beckham et al. recommended 125° knee and 145° hip in the study you referenced. But you stated it the other way round “initial hip and 146 knee angles were approximately 125° and 145°, respectively.” Why is that so?
Author Response: Thank you for bringing this to our attention. This was a typo and meant to reflect the suggested angles outlined by Beckham et al. This has now been updated in the Methods.
The current data included 4 females. Females generally have different body fat percentage as men. Hence, the BMI reading may not be applicable to men, and vice versa. Would the authors be able to provide another set of data that exclude the female participants? As a table in the supplementary document at least.
Author Response: We agree in that there are known sex differences for anthropometrics and body composition. Unfortunately, we did not have control over the number of females participating in the study and did not have equal distribution to allow for any comparisons by sex. However, similar to the military, the fire fighting profession does not adjust fitness standards, expectations of occupational performance or work activities based upon sex and therefore including both men and women together maintains ecological validity as to what is common practice in the field.
The current sample size of female participants is too small to allow for this data to be generalized for the female as well. Should state this in the limitations.
Author Response: We have now added this to the limitation.
Reviewer 2 Report
The article propose a specific sequence of exercises in an air management courses (AMC) as a method to provide a measure for ability to assess work efficiency in order to characterize occupational performance in the firefighters’ occupation and try to use this Assessment Center (AC) type approach to check correlation of the result achieved by the AC’s participant with dichotomized BMI index score.
This interpretation of the article is obviously unfair and corresponds primarily to its title, since even the abstract is much more nuanced. However, attention is drawn to this title and this result, which - if taken seriously - would provide a useful tool for cheap pre-selection in employment when recruiting new people to the firefighter profession, as well as a methodology for recognizing people practicing this profession, which should be subjected to the necessary practical tests.
Such a radical conclusion can be drawn due to the insufficient description of the limitations of the study, but this conclusion seems to be in line with the intentions of the authors of the article in the version received by the reviewer.
Hence, it is worth emphasizing clearly that their research does not justify such a conclusion in any way, which must be clearly and firmly written in the conclusions of the text.
First, each AC as a set of exercises - which the authors rightly emphasize (87-88 "a challenge with AMC's and similar occupational performance assessments, is they are often not standardized; thereby, making assessment of physiological demands, and characterization of performance outcomes challenging" ) - to be useful as a direct measure of professional suitability, it would have to be standardized. In this case, standardization means setting weights for individual tasks that constitute a burden in a specific type of professional work (simulated or real “critical incidents”), so that they correspond to the weight that a given type of work has in the actual performance of a professional role.
Secondly, which the authors seem to be aware of (describing the exercise sequences in detail in lines 175-203) - especially in the case of physical activity (but not only), the order of the exercises performed (sequence) is important both because of their mutual interaction during implementation and due to the prognostic (ecological) accuracy of predicting performance in a natural situation (similarity to a typical and important natural task).
Thirdly, a study focused on physical fitness and efficiency in practice ignores other qualities of the performer of the work, which - with a minimum level of this fitness - make the performer a valuable team member. However, research on fitness itself does not provide a measure of either this minimum level of fitness per se, or the minimum level of fitness for individual exercises, whether performed as a separate exercise or as a set of exercises (sequence).
The authors clearly point out that "Standard laboratory measures and protocols are common for evaluation of health and fitness parameters such as aerobic capacity, cardiovascular function, and body composition. Such tests may lack ecological validity and may not be accessible to all firefighters" (lines 77-81) but at the same time they seem to believe that it is possible - without a QUANTITATIVE analysis of firefighting jobs - to create "The development of a field-based measure of work efficiency can quantify occupational performance and facilitate the compilation of rankings for individual firefighters and provide normative data. Further, a singular measure of work efficiency would permit the examination of relationships between fitness parameters and occupation performance” (lines 110-114), so they set themselves the goal of “developed field-based [in what sense field?? they want to create it based on a simulation, not a FIELD/JOB] measure of work efficiency in order to characterize occupational performance among firefighters during an open-ended task.” (lines 116-118). Such a goal suggests the intention to create a set of exercises that are adequate for the ENTIRE PROFESSION with characteristic critical incidents with a fixed mutual weight (theoretically and a priori) - which seems to be obviously impossible to perform.
The presented study has a number of other limitations (or real problems), apart from the adequacy of the presented set of exercises as a simulated work samples (or more broadly - as a simulated professional suitability test, as the title of the article suggests such use).
The analyzed sample consists of only 57 people and the assumption of the minimum sample size for the number of predictors used is not met (at least 15 per one analyzed predictor).
The small sample exludes an adequate use of BMI index. The dichotomization of this variable causes that the text evokes resistance in the reader, and its implementation may raise reader's doubts. BMI was not invented to measure the fitness of an individual person, but as a measure for the population, because it does not take into account individual factors related to both body structure and age. Introduction to the article, which suggests that BMI is a problem among firefighters (lines 63-65: “Previous research among a cohort of nearly 500 career firefighters reported that 80% of firefighters were classified as overweight (Body mass index [BMI] > 25 kg m-2 ) and 34% classified as obese (BMI > 30 kg m-2 )") suggest treating this measure as a pre-selection indicator or used to assess the professional suitability of the contractor, which is misleading and may raise legal doubts. The belief hidden behind such use that there is a causal relationship between BMI (and – in the text the dichotomous BMI) and fitness or physical capacity does not seem to have a scientific justification, not only at present, but it should not be possible to obtain such a justification at all (due to factors which BMI does not measure).
Summing up this element: in the limitations of the conducted study, it should be clearly stated that an ordinal BMI (multileveled) measurement is a necessary as a supplement (or a future) study, to give a possibility to formulate suggestions of similar generality as those formulated by the authors.
Thirdly - and the omission of this element is a surprise for the reviewer - the study should be supplemented by an attempt to measure the accuracy of the obtained indicator (or results of Assessment Center measurements), whether by measuring the opinions of colleagues about the professional fitness of the respondents, or by using other measures of the actual professional usefulness of the firemen for whom the AMC way the measurement of their fitness and physical capacity were conducted.
Any attempt to prepare a reasonable AC requires the use of data on practical functioning in a professional role, and not only measurements in a simulated situation - the authors had access to the subjects (firemen) and their professional environment, so the reviewer is surprised that they did not used these data sources. At a minimum, t it must be indicated as a necessary direction for further research supplementing this very preliminary idea for a research program.
In other words - the title of the article must emphasize the preliminary and hypothetical value of the obtained result.
Author Response
Reviewer 2:
The article propose a specific sequence of exercises in an air management courses (AMC) as a method to provide a measure for ability to assess work efficiency in order to characterize occupational performance in the firefighters’ occupation and try to use this Assessment Center (AC) type approach to check correlation of the result achieved by the AC’s participant with dichotomized BMI index score.
This interpretation of the article is obviously unfair and corresponds primarily to its title, since even the abstract is much more nuanced. However, attention is drawn to this title and this result, which - if taken seriously - would provide a useful tool for cheap pre-selection in employment when recruiting new people to the firefighter profession, as well as a methodology for recognizing people practicing this profession, which should be subjected to the necessary practical tests.
Author Response: The title has been changed to remove any unintentional bias.
Such a radical conclusion can be drawn due to the insufficient description of the limitations of the study, but this conclusion seems to be in line with the intentions of the authors of the article in the version received by the reviewer. Hence, it is worth emphasizing clearly that their research does not justify such a conclusion in any way, which must be clearly and firmly written in the conclusions of the text.
Author Response: Thank you for the response. We have now acknowledged some of these limitations in the Discussion section and have also decided to change the title of the manuscript so that it better reflects findings from the manuscript, without concerns of the limitations precluding an appropriate conclusion.
First, each AC as a set of exercises - which the authors rightly emphasize (87-88 "a challenge with AMC's and similar occupational performance assessments, is they are often not standardized; thereby, making assessment of physiological demands, and characterization of performance outcomes challenging" ) - to be useful as a direct measure of professional suitability, it would have to be standardized. In this case, standardization means setting weights for individual tasks that constitute a burden in a specific type of professional work (simulated or real “critical incidents”), so that they correspond to the weight that a given type of work has in the actual performance of a professional role.
Author Response: Thank you for the feedback. While directly comparing results from one proprietary AMC to another (such as from another department) may not be appropriate because of differences in course structure, exercise order etc., the work efficiency formula could be used for any AMC (assuming access to SCBA, timing devices, and accelerometry) to produce a computed score for an AMC. This would still allow for the compilation of departmental rankings, evaluate occupational performance, and evaluate progress over time.
Secondly, which the authors seem to be aware of (describing the exercise sequences in detail in lines 175-203) - especially in the case of physical activity (but not only), the order of the exercises performed (sequence) is important both because of their mutual interaction during implementation and due to the prognostic (ecological) accuracy of predicting performance in a natural situation (similarity to a typical and important natural task).
Author Response: Assuming the order of exercises is consistent from one AMC to the next (in subsequent evaluations or years of completing the AMC), the computed measure of work efficiency would allow for improvements to be observed if the fitness level of firefighters were to improve over time.
Thirdly, a study focused on physical fitness and efficiency in practice ignores other qualities of the performer of the work, which - with a minimum level of this fitness - make the performer a valuable team member. However, research on fitness itself does not provide a measure of either this minimum level of fitness per se, or the minimum level of fitness for individual exercises, whether performed as a separate exercise or as a set of exercises (sequence).
Author Response: Thank you for the comment. – an interesting point.
The authors clearly point out that "Standard laboratory measures and protocols are common for evaluation of health and fitness parameters such as aerobic capacity, cardiovascular function, and body composition. Such tests may lack ecological validity and may not be accessible to all firefighters" (lines 77-81) but at the same time they seem to believe that it is possible - without a QUANTITATIVE analysis of firefighting jobs - to create "The development of a field-based measure of work efficiency can quantify occupational performance and facilitate the compilation of rankings for individual firefighters and provide normative data. Further, a singular measure of work efficiency would permit the examination of relationships between fitness parameters and occupation performance” (lines 110-114), so they set themselves the goal of “developed field-based [in what sense field?? they want to create it based on a simulation, not a FIELD/JOB] measure of work efficiency in order to characterize occupational performance among firefighters during an open-ended task.” (lines 116-118). Such a goal suggests the intention to create a set of exercises that are adequate for the ENTIRE PROFESSION with characteristic critical incidents with a fixed mutual weight (theoretically and a priori) - which seems to be obviously impossible to perform.
Author Response: In the context of exercise and sport science settings, the term “field” refers to any tests or evaluations performed outside of a controlled laboratory environment. It does not necessarily refer to live-action or occurring during an active emergency rescue situation. Further, the term work efficiency is modeled after the term exercise economy which describes the physiological and metabolic efficiency of the human body during exercise. However, because firefighting is an occupation, we chose to use the term work efficiency to better reflect the task at hand and what had been done previously by Norris et al. 2021.
The presented study has a number of other limitations (or real problems), apart from the adequacy of the presented set of exercises as a simulated work samples (or more broadly - as a simulated professional suitability test, as the title of the article suggests such use).
Author Response: Thank you for the comment. We have addressed your concerns and suggestions below, namely changing the title of the manuscript and adding several limitations to the end of the Discussion.
The analyzed sample consists of only 57 people and the assumption of the minimum sample size for the number of predictors used is not met (at least 15 per one analyzed predictor).
Author Response: Excellent point. To clarify, we used simple linear regression to examine the extent to which the variables individually predicted work efficiency. We did not use multiple linear regression to combine the variables into a multivariate model, due to our own concerns about sample size, as well as potential multicollinearity among the variables. We have revised the manuscript to clarify (page 6; line 230). The minimum sample size per predictor variable guidelines (e.g., 15 subjects per predictor variable) are generally intended to be applied with multivariate models. Our sample size should be sufficient for simple linear regression. Thank you for allowing us to clarify.
The small sample excludes an adequate use of BMI index. The dichotomization of this variable causes that the text evokes resistance in the reader, and its implementation may raise reader's doubts. BMI was not invented to measure the fitness of an individual person, but as a measure for the population, because it does not take into account individual factors related to both body structure and age. Introduction to the article, which suggests that BMI is a problem among firefighters (lines 63-65: “Previous research among a cohort of nearly 500 career firefighters reported that 80% of firefighters were classified as overweight (Body mass index [BMI] > 25 kg m-2 ) and 34% classified as obese (BMI > 30 kg m-2 )") suggest treating this measure as a pre-selection indicator or used to assess the professional suitability of the contractor, which is misleading and may raise legal doubts. The belief hidden behind such use that there is a causal relationship between BMI (and – in the text the dichotomous BMI) and fitness or physical capacity does not seem to have a scientific justification, not only at present, but it should not be possible to obtain such a justification at all (due to factors which BMI does not measure).
Author Response: We agree in that BMI does not account for specific tissue and composition related differences. However, the way BMI is used to categorize weight, relative to height, and classify individuals as normal weight, overweight or obese is commonly used (despite its limitations). Further, a course such as the AMC, may be biased towards smaller, lighter individuals who may require a lower amount of air to do a fixed amount of work. Therefore, examining differences in the physiological responses to the AMC between weight categories provides preliminary insight into how such a course may be more “challenging’ for larger statured individuals, regardless of compositional differences.
Summing up this element: in the limitations of the conducted study, it should be clearly stated that an ordinal BMI (multileveled) measurement is a necessary as a supplement (or a future) study, to give a possibility to formulate suggestions of similar generality as those formulated by the authors.
Thirdly - and the omission of this element is a surprise for the reviewer - the study should be supplemented by an attempt to measure the accuracy of the obtained indicator (or results of Assessment Center measurements), whether by measuring the opinions of colleagues about the professional fitness of the respondents, or by using other measures of the actual professional usefulness of the firemen for whom the AMC way the measurement of their fitness and physical capacity were conducted.
Any attempt to prepare a reasonable AC requires the use of data on practical functioning in a professional role, and not only measurements in a simulated situation - the authors had access to the subjects (firemen) and their professional environment, so the reviewer is surprised that they did not used these data sources.
Author Response: We are unsure of what you mean regarding “professional role” or AC. While the AMC is a simulation, it consists of several fire fighter specific tasks regularly used in training exercises. It is not reasonable to conduct research of this nature during live-action fire rescue activities because of the concerns of impeding or causing a distraction to first responders involved in a life-or-death situation.
At a minimum, it must be indicated as a necessary direction for further research supplementing this very preliminary idea for a research program. In other words - the title of the article must emphasize the preliminary and hypothetical value of the obtained result.
Author Response: With the revised title of the manuscript, we feel as though it is a better representation of the study’s focus and primary findings. We have added commentary to the end of the Discussion providing directions for future research with a larger sample size.
Round 2
Reviewer 2 Report
Thank you for your answers and comments. My field of specialite is HR and I have feeling that this point of view is missed in your text or - better - you not see practical consequences of the statement you put in the text, without proper limmitation.
I would stress as a limitation that small sample do not make possible to inquaire BMI as a ordinal variable, as well as that the phisical fitness is an only one from many qualities which are important for in-the-job perfrmance of the firefither, which importance in the whole performance has not been assess in your study, as well as - it is possible that only a specific level of this fitness is a nessecary condition for the job. Assessment Center (and your study is about this tool in development and selection area of HRM) is appropiate for selection ONLY in the context of competency analysis of the job - and you had not these competencies which are usefull for profession (only one - an important one - aspect of their duties).
Author Response
Thank you for your answers and comments. My field of specialite is HR and I have feeling that this point of view is missed in your text or - better - you not see practical consequences of the statement you put in the text, without proper limmitation.
Author Response: Thank you for again taking the time to review our revised manuscript. We have addressed your concerns below and have added to the limitation sections to highlight these issues.
I would stress as a limitation that small sample do not make possible to inquaire BMI as a ordinal variable, as well as that the phisical fitness is an only one from many qualities which are important for in-the-job perfrmance of the firefither, which importance in the whole performance has not been assess in your study, as well as - it is possible that only a specific level of this fitness is a nessecary condition for the job. Assessment Center (and your study is about this tool in development and selection area of HRM) is appropiate for selection ONLY in the context of competency analysis of the job - and you had not these competencies which are usefull for profession (only one - an important one - aspect of their duties).
Author Response: Using BMI as a categorical factor to assess differences in physiological responses and work efficiency was not a primary aim of the current study, nor was it used to conduct an a priori power analysis. However, when completing a power analysis, if a goal of detecting an effect size of 0.8 (large) in regard to differences in work efficiency was identified using a between-group design, a total sample size of 52 subjects (26 in each group) would be required for 80% power and an alpha of 0.05. Our current sample totalled 57. However, there was an unequal distribution across groups (n=21 and n=36 in the group with a BMI < 25 kg·m-2, and > 25 kg·m-2, respectively). Despite the unequal distribution, we feel as though it was still appropriate to stratify our sample based upon this criterion. Further, we have acknowledged that a simple linear regression analysis was used, and the current sample size is sufficient as we did not perform multiple linear regression to combine the variables into a multivariate model.
We have added information to the Discussion (limitations) section to reiterate that we only evaluated a select number of fitness characteristics and only assessed occupational performance using a single test. We agree in that it is important to note that other fitness characteristics may play a role in occupational performance, which also is more than a single test and other aspects of the profession or firefighter abilities are also an important consideration (calm under pressure, decision making, tactical skills, knowledge of fires, etc.).
